# Study designs for clinical trials applied to personalised medicine: a scoping review

Cecilia Superchi ![ORCID],[1] Florie Brion Bouvier ![ORCID],[1] Chiara Gerardi ![ORCID],[2] Montserrat Carmona ![ORCID],[3,4] Lorena San Miguel ![ORCID],[5] Luis María Sánchez-Gómez ![ORCID],[3,4] Iñaki Imaz-Iglesia ![ORCID],[3,4] Paula Garcia,[6] Jacques Demotes ![ORCID],[6] Rita Banzi ![ORCID],[2] Raphaël Porcher ![ORCID],[1] the PERMIT Group

¹Centre of Research in Epidemiology and Statistics, Université de Paris, Paris, Île-de-France, France
²Center for Health Regulatory Policies, Istituto di Ricerche Farmacologiche Mario Negri, Milano, Lombardia, Italy
³Agencia de Evaluación de Tecnologias Sanitarias, Instituto de Salud Carlos III, Madrid, Spain
⁴Red de Investigación en Servicios de Salud en Enfermedades Crónicas (REDISSEC), Madrid, Spain
⁵Belgian Health Care Knowledge Centre (KCE), Brussels, Belgium
⁶European Clinical Research Infrastructure Network (ECRIN), Paris, France

**Correspondence to**
Dr Cecilia Superchi;
cecilia.superchi@gmail.com

## ABSTRACT

**Objective** Personalised medicine (PM) allows treating patients based on their individual demographic, genomic or biological characteristics for tailoring the 'right treatment for the right person at the right time'. Robust methodology is required for PM clinical trials, to correctly identify groups of participants and treatments. As an initial step for the development of new recommendations on trial designs for PM, we aimed to present an overview of the study designs that have been used in this field.

**Design** Scoping review.

**Methods** We searched (April 2020) PubMed, Embase and the Cochrane Library for all reports in English, French, German, Italian and Spanish, describing study designs for clinical trials applied to PM. Study selection and data extraction were performed in duplicate resolving disagreements by consensus or by involving a third expert reviewer. We extracted information on the characteristics of trial designs and examples of current applications of these approaches. The extracted information was used to generate a new classification of trial designs for PM.

**Results** We identified 21 trial designs, 10 subtypes and 30 variations of trial designs applied to PM, which we classified into four core categories (namely, Master protocol, Randomise-all, Biomarker strategy and Enrichment). We found 131 clinical trials using these designs, of which the great majority were master protocols (86/131, 65.6%). Most of the trials were phase II studies (75/131, 57.2%) in the field of oncology (113/131, 86.3%). We identified 34 main features of trial designs regarding different aspects (eg, framework, control group, randomisation). The four core categories and 34 features were merged into a double-entry table to create a new classification of trial designs for PM.

**Conclusions** A variety of trial designs exists and is applied to PM. A new classification of trial designs is proposed to help readers to navigate the complex field of PM clinical trials.

## INTRODUCTION

Personalised medicine (PM) is an evolving field, which allows treating patients by providing them a specific therapy according to their individual demographic, genomic or biological characteristics.[1] It was defined by

## STRENGTHS AND LIMITATIONS OF THIS STUDY

⇒ This is the first review, which systematically searched for all trial designs applied to personalised medicine.
⇒ The screening process and data extraction were performed in duplicate.
⇒ A new classification of trial designs for personalised medicine has been proposed.
⇒ We cannot exclude that we missed some relevant designs since we restricted the search to the last 15 years.

the European Council Conclusion on PM as 'a medical model using characterisation of individuals' phenotypes and genotypes (eg, molecular profiling, medical imaging, lifestyle data) for tailoring the right therapeutic strategy for the right person at the right time, and/or to determine the predisposition to disease and/or to deliver timely and targeted prevention'.[2]

Many trial designs have been used to evaluate personalised treatment or interventions.[3] The most common design is the enrichment design, whereby only biomarker-positive patients are randomly assigned to the targeted or control arm.[4] Despite its popularity, the use of enrichment designs is recommended only when the biomarker is a perfect predictor of the response in order not to deny biomarker-negative patients a treatment they would have otherwise benefited from.[5] Prospective validation of the candidate biomarker is therefore strongly recommended before applying these trials designs.

Over the last years, more complex study designs have been increasingly proposed in the field of PM.[4] According to the Clinical Trials Facilitation and Coordination Group, a clinical trial is considered as using a complex design 'if it has separate parts that could

constitute individual clinical trials and/or is characterised by extensive prospective adaptations such as planned additions of new Investigational Medicinal Products or new target populations'.[6] These designs are particularly efficient because they allow answering multiple clinical research questions within a single study.[7] Examples of common complex designs are the so-called basket, umbrella and platform trials, which are frequently applied in the field of oncology.[8] Basket trials refer to designs in which patients with heterogeneous diagnoses but with similar disease mechanisms are tested using the same targeted therapy. While, umbrella trials evaluate multiple treatment options in patient groups, which present the same disease, but with different genetic mutations. Finally, platform trials allow testing multiple targeted therapies in patients with the same disease in a perpetual manner, using interim evaluations and allowing therapies to enter or leave the trial.[9] However, these designs are often challenging[6] because they often require independent statistical analyses for each subprotocol, including interim analyses driving prospective adaptation with the addition of new interventions or populations, and/or termination of subprotocols based on futility or safety issues.

Numerous methodological challenges, covering many aspects of the study design (eg, randomisation, use of control arm, biomarker stratification, biomarker validation), are associated with trial designs applied to PM. The application of robust methodologies is especially important for clinical trials applied to PM to correctly select participants and treatments to be tested. As a starting point for the development of new recommendations on the use of trial designs applied to PM, we aimed to map the landscape of the existing study designs for clinical trials applied to this medical field.

Our specific objectives were to answer to the following five research questions:

1. What are the available designs for clinical trials applied to PM?
2. What are the examples of current applications of these approaches?
3. What are the pros and cons of the different approaches?
4. How is a PM strategy versus non-personalised strategy evaluated?
5. What are the gaps in the current research on PM clinical trials?

This scoping review is part of the PERMIT project (PERsonalised MedIcine Trials) aimed at mapping the methods for PM research and building recommendations on robustness and reproducibility of different stages of the development programmes. Although several categorisation may be proposed, the PERMIT project considers four main building blocks of the PM research pipeline: (1) design, building and management of stratification and validation cohorts; (2) application of machine learning methods for patient stratification; (3) use of preclinical methods for translational development, including the use of preclinical models used to assign treatments to patient clusters; (4) evaluation of treatments in randomised clinical trials. This scoping review covers the fourth building block in this framework.

## METHODS

We conducted a scoping review following the methodological framework suggested by the Joanna Briggs Institute.[10] The framework consists of six stages: (1) identifying the research questions, (2) identifying relevant studies, (3) selecting the studies, (4) charting the data, (5) collating, summarising and reporting results and (6) pursuing a consultation.

A study protocol was published in Zenodo before conducting the review.[11] Due to the iterative nature of scoping reviews, deviations from the protocol were expected and duly reported when occurred. We used the PRISMA-ScR (Preferred Reporting Items for Systematic reviews and Meta-Analyses extension for Scoping Reviews) checklist to report our results.[12]

### Study identification

Relevant studies and documents were identified balancing feasibility with breadth and comprehensiveness of searches. We searched PubMed, Embase and the Cochrane Library (search date: 7–8 April 2020) for all reports describing a study design for clinical trials applied to PM. Online supplemental file 1 reports the search strategies applied. We did not restrict the search to any publication type. Because many systematic and narrative reviews on trial designs applied to PM have already been published over the last years, we limited our search from 2005 to April 2020. We restricted inclusion to English, French, German Italian and Spanish languages. We searched for the grey literature on websites of existing projects about innovative clinical trials (eg, EU-PEARL) and by consulting partners of the PERMIT project.

### Eligibility criteria and deviation from the protocol

We included all reports describing a trial design applied to PM. The operational definition of PM used in the present study is reported in box 1. Because of the extensive volume of literature related to trial designs in PM, we restricted the inclusion criteria to trial designs for phase II, III and IV. We excluded single-arm trials, which are not part of a master protocol, non-adaptive enrichment design and N-of-1 trials. We also excluded publications such as prefaces to a special issue and speaker, symposium and panel abstracts, posters and letters to the editor due to the limited information usually provided. These exclusion criteria were not specified in the protocol, but they were agreed among the authors before starting the screening process. The research question 'What are the pros and cons of the different approaches?' (ie, objective 3) is not reported in the present paper, and will be subject to a specific study.

### Study selection

We exported the references retrieved from the searches into the Rayyan online tool.[13] Duplicates were removed

---

**Box 1    Personalised medicine definition**

**What is Personalised Medicine?**
According to the European Council Conclusion on personalised medicine for patients personalised medicine is 'a medical model using characterisation of individuals' phenotypes and genotypes (eg, molecular profiling, medical imaging, lifestyle data) for tailoring the right therapeutic strategy for the right person at the right time, and/or to determine the predisposition to disease and/or to deliver timely and targeted prevention'.[2]

In the context of the PERMIT project, we applied the following common operational definition of personalised medicine research: a set of comprehensive methods, (methodological, statistical, validation or technologies) to be applied in the different phases of the development of a personalised approach to treatment, diagnosis, prognosis or risk prediction. Ideally, robust and reproducible methods should cover all the steps between the generation of the hypothesis (eg, a given stratum of patients could better respond to a treatment), its validation and preclinical development and up to the definition of its value in a clinical setting.[11]

---

automatically using the reference manager EndNote V.X9 (Clarivate Analytics, Philadelphia, USA) and manually by one author (CS). Eligible reports applying a particular trial design were retrieved from the search strategies and screened by reviewers. Five reviewers independently screened the titles and abstracts: one reviewer (CS) screened all the records and four reviewers (II-I, LMS-G, LSM and PJ) screened 25% of references each. Due to the involvement of many reviewers, we conducted a pilot screening using 56 articles (2.5%), corresponding to the articles published from 1 January 2020 to search date (7–8 April 2020), to verify whether all reviewers used the same inclusion and exclusion criteria. We retrieved full-text copies of potentially eligible reports for further assessment. Six reviewers independently confirmed the eligibility: one reviewer (CS) examined all full-text copies and five reviewers (IB, II-I, LMS-G, MMPS and SLM) assessed 20% of references each. Disagreements were solved by consensus or by involving a third expert reviewer (RP).

### Charting the data
We designed a data extraction form using Google Forms (online supplemental file 1). General study characteristics extracted were as follows: first author name, title of article, contact detail of corresponding author, year of publication and type of publication. In addition, for each trial design referred to in the paper, we collected information on its definition, methodology, statistical considerations, advantages, disadvantages, utility, gaps and examples of actual trials, which adopted the design. A list of trial designs, which were retrieved from two previously conducted systematic reviews,[14 15] was included in the data extraction form to harmonise the names used to report the same trial design. This initial list of trial designs was used as starting point to classify the identified trial designs and then modified and expanded on based on the results obtained in the present scoping review.

When the trial design name reported in the paper did not match any of the trial design names included in the list, reviewers recorded the trial name verbatim.

Two reviewers (CS and FBB) piloted and refined the data extraction form using three reviews (4%). Since many narrative reviews were already published about trial designs applied to PM, the data extraction was conducted in two phases. First, two reviewers (CS and FBB) independently extracted data from the identified systematic and narrative reviews. Second, three reviewers (CS, FBB and MC) working independently extracted data for all the remaining selected records, which were neither a systematic nor narrative review, only if they provided new information, which was not extracted in the previous phase. One reviewer (FBB) extracted data from all records and two reviewers (CS and MC) extracted 60% and 40% of articles, respectively. Differences in terminology were discussed between reviewers to ensure that the same trial designs were included in the same category. Disagreements were solved by consensus or by involving a third expert reviewer (RP).

It was not within the remit of this scoping review to assess the methodological quality of individual studies included in the analysis.

### Collating, summarising and reporting results
We summarised the extracted data in tables and figures. Information on the definition, methodology, statistical considerations, advantages, disadvantages, utility and gaps of trial designs was extracted verbatim. Data on the examples of clinical trials adopting the different approaches were summarised using frequencies and percentages.

A researcher (CS) listed all study designs and identified the central feature(s) for each of them, which were grouped into feature domains. The initial list was reviewed by a senior statistician with expertise in designing clinical trials (RP). A final list was created and agreed on with members of the PERMIT steering committee and coauthors of the present study. The list of features was therefore based on the identified study designs and also the expertise of members of the PERMIT project.

### New classification of trial designs in PM
Based on the identified trial designs and features, we proposed a new classification of trial designs for PM. Other attempts in classifying trial designs applied to PM have been proposed in the literature. However, they were limited to classifying the designs into categories[3 4 8] or identifying the design based on a specific feature (eg, adaptive or non-adaptive trials).[14 15] This new classification goes a step further, proposing a new approach in classifying the trial designs considering two variables, which are core designs and design features, into a double-entry table.

### Consultation exercise
The members of the PERMIT consortium, associated partners and the PERMIT project Scientific Advisory

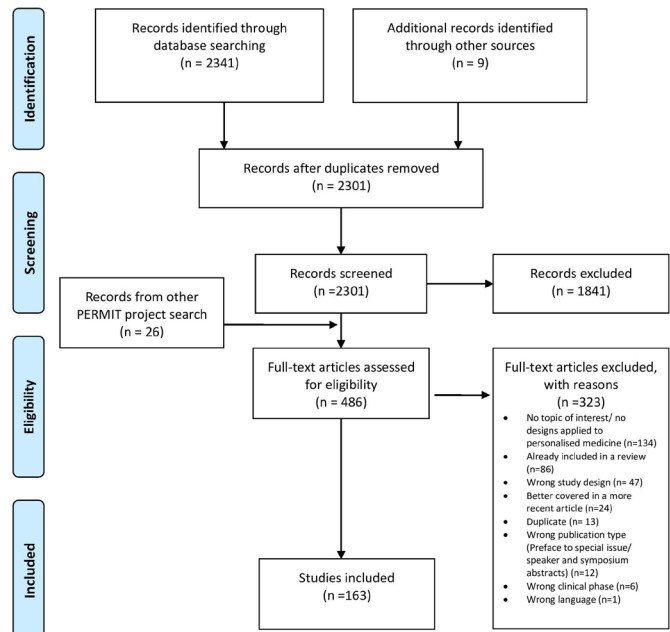

**Figure 1** Study selection flow diagram. PERMIT, PERsonalised MedIcine Trials.

Board discussed the preliminary findings of the scoping review in a 2-hour online workshop. A first version of the classification of the trial designs in PM was presented and discussed.

### Patient and public involvement

The European Patients Forum is a member of PERMIT project. Although not directly involved in the conduction of the scoping review, they received the draft review protocol for collecting comments and feedback.

### RESULTS

### Study selection and general characteristics of reports

We retrieved 2350 citations from the electronic search and after removing the duplicates, 2301 remained. We excluded 1841 records based on titles and abstracts. After full-text assessment, 323 publications were excluded, and 163 met the inclusion criteria (see flow chart in figure 1 and online supplemental file 1; the data extraction including information on the general study characteristics and definition, methodology, statistical considerations and examples of each study design referred to in each included paper, is available on the online platform Zenodo[16]). From these 163 publications, we identified 5 systematic reviews, 66 narrative reviews, 8 original research articles, 26 methodological studies, 4 study protocols, 37 conference abstracts, 4 commentaries, 2 discussion papers, 3 reports, 1 book chapter, 1 editorial, 1 guidance document and 5 links about trial registration (eg, ClinicalTrials.gov).

### Trial designs and core designs in PM

We identified 21 trial designs, 10 subtypes and 30 variations of trial designs applied to PM (online supplemental file 1). Information on the definition, methodology and statistical considerations of identified trial designs are reported on the online supplemental file 1.

We classified the trial designs into four core categories named as *Master protocols*, *Randomise-all*, *Biomarker-strategy* and *Enrichment*. Building on the definitions provided by Tajik *et al*[3] and Park *et al*,[8] we defined the four core categories as:

▶ *Master protocols*: trial design, which includes multiple parallel substudies under a common infrastructure.

▶ *Randomise-all*: trial design where patients meeting the eligibility criteria, irrespective of their biomarker status, are randomised to either an experimental or control treatment. This category also includes those hybrid designs, which first use a *Randomise-all* design, and then only a specific biomarker defined subgroup is randomised to either an experimental or control treatment.

▶ *Biomarker-strategy*: trial design where eligible patients are randomised to either a marker-based treatment strategy or non-marker-based treatment strategy.

▶ *Enrichment*: trial design where eligibility is determined according to the biomarker status and patients are then randomised to either an experimental or control treatment. A specific biomarker defined subgroup (usually biomarker positives) is believed to benefit more from a treatment compared with the other subgroup (usually biomarker negatives).

An example of a study design for each core category, including its definition and methodology used, is shown in table 1. Overall, we identified 5 trial designs, 6 subtypes and 7 variations for *Master protocols*, and 10 trial designs, 2 subtypes and 22 variations for *Randomise-all*, 5 trial designs for *Biomarker-strategy* and 1 trial design, 2 subtypes and 1 variation for *Enrichment*.

From the identified designs, we found 34 main features of trial designs in PM, which were clustered into 11 features domains (table 2). The feature domains include the key design features that characterise a trial design for PM such as framework, model, control group, randomisation, biomarker assessment and adaptive aspects, and that should be carefully considered when designing a trial. A new classification of the trials designs for PM has been proposed and is reported in table 3. The classification is presented in a double-entry table, which includes the main trial features on the y-axis and core categories of the trial designs on the x-axis.

### General characteristics of clinical trials in PM

We found 131 clinical trials, which used the identified designs (online supplemental file 1). Table 4 presents the general characteristics of the identified trials.

Most trials used a basket (35/131, 26.7%), umbrella (30/131, 22.9%), platform (18/131, 13.7%) or marker stratified (15/131, 11.5%) design. The great majority of the trials were in the field of oncology (113/131, 86.3%). At the time of writing (March 2021), the recruitment status was ongoing for 48.1% (63/131) of the trials. A trial

**Table 1**  Examples of core categories

| Core category | Study design example | Study design definition | Study design methodology |
|---|---|---|---|
| Master protocols | Platform | 'A platform trial is a single histology randomized phase II clinical trial involving multiple biomarkers and multiple drugs. Rather than assuming that we know which drug is appropriate for which biomarker stratum, randomization among drugs is used in the platform trial.'[43] | 'Initially the treatments are randomized with equal weights to the patients of a stratum. As data accumulates, the randomization weights change to favour assignment of drugs with higher within-stratum response rates. The endpoint used must be observed early enough to enable adaption of randomization weights.' [43] |
| Randomise-all | Biomarker-positive and overall strategies with fall-back analysis | 'It evaluates both the treatment effect in the overall study population and in the biomarker-positive subgroup sequentially.'[14] | 'In the fall-back design, we first test the overall population using the reduced significance level $\alpha_1$ and if the test is significant, we consider that the novel treatment is effective in the overall population; however, if the result is not significant then we test the treatment effect in the biomarker-positive subgroup using the level of significance $\alpha_2 = \alpha - \alpha_1$, where $\alpha$ is the overall significance level (Type I error rate). The significance levels $\alpha$ can be considered as one-sided or two-sided significance levels.'[14] |
| Biomarker strategy | Biomarker-strategy design with treatment randomisation in the control arm | 'The biomarker-strategy design with treatment randomization in the control treatment is able to inform us about whether the biomarker-based strategy is better than not only the standard treatment but also better than the experimental treatment in the overall population.' [14] | 'Patients are first randomly assigned to either the biomarker-based strategy arm or to the non-biomarker-based strategy arm. Next, patients who are allocated to the non-biomarker-based strategy are again randomized either to the experimental treatment arm or to the standard treatment arm irrespective of their biomarker status. Patients who are allocated to the biomarker-based strategy and who are biomarker-positive are given the experimental treatment and patients who are biomarker-negative are given the control treatment.' [14] |
| Enrichment | Adaptive threshold sample-enrichment design | 'It is a two-stage design in a Phase III setting(...) to adaptively modify accrual in order to broaden the targeted patient population.' [15] | 'At the interim analysis stage, the treatment effect of a sample of patients ($n_1$) from the biomarker-positive subset is estimated. If an improvement is seen in the experimental treatment arm which is greater than a pre-specified threshold value (ie, the estimated treatment difference between the novel treatment arm and the control treatment arm for this subpopulation is greater than a threshold value c divided by the square root of the aforementioned sample size $n_1$) the trial continues with accrual of patients from the entire biomarker-positive subgroup and additional patients are also accrued from the biomarker-negative subpopulation; otherwise the trial is stopped for futility. At the end of the trial, the treatment effect is estimated for all subpopulations. Researchers should choose the sample size $n_1$ so that a persuasive result can be reached when the first stage of the trial is completed.' [15] |

**Table 2** Main features of trial designs applied to personalised medicine

| Feature domains | Features |
|---|---|
| Inference framework | Bayesian |
| | Frequentist |
| Model* | Disease progression* |
| | Longitudinal* |
| | Hierarchical |
| Control group | Common/shared† |
| | Contemporaneous‡ |
| | Historical§ |
| Randomisation | With treatment randomisation in both biomarker-positive and biomarker-negative subgroups |
| | Without treatment randomisation in the biomarker-negative subgroup¶ |
| | Only for patients with discordant clinical and genomic risk evaluation** |
| Randomisation in the non-biomarker based strategy arm | With treatment randomisation |
| | Without treatment randomisation†† |
| | Reverse biomarker strategy‡‡ |
| Subgroup specific | Sequential subgroup specific§§ |
| | Parallel subgroup specific¶¶ |
| Biomarker positive and overall strategies*** | With sequential assessment |
| | With parallel assessment |
| | With fall-back analysis††† |
| | Marker sequential test‡‡‡ |
| Biomarker assessment | With biomarker assessment in the entire population |
| | Without biomarker assessment in the control arm |
| Personalised medicine (PM) specific adaptive aspects§§§ | Adaptive enrichment |
| | Adaptive signature |
| | Threshold determination¶¶¶ |
| Generic adaptive aspects | Adding a new arm |
| | Early stopping**** |
| | Interim analysis†††† |
| | Outcome-based adaptive randomisation |
| | Sample size reassessment |
| | Seamless |
| Treatment tailoring aspects | Pharmacodynamic biomarker assessment after run-in phase period‡‡‡‡ |
| | Dynamic treatment regime§§§§ |
| | PK/PDmodelling¶¶¶¶ |

*Model used for analysis. A disease progression model takes into account the patient disease state and other patient baseline characteristics for charactering patient clinical outcome(s).[44] Longitudinal model permits including in the analysis the partial information of patients who have not yet reached their final outcome at an interim analysis.[44]

†A common/shared control group can be used in a trial design in which multiple treatments are being tested, instead of each treatment having its own control arm.

‡If patients in the common/shared control group receive a 'Standard of care' that may change over time or the profile of the patients enrolled on the trial may change over time, a trial design can use a contemporaneous control group meaning that the comparison of treatment's effects may be restricted to those patients who were enrolled/randomised in the same period as those patients who were allocated to the treatment.

§If a comparison group is not available in the existing trial or substudy or at the same time but in a different setting, a trial design can use a historical control consisted of a group of individuals treated in the past.

¶Patients in the biomarker-negative subgroup receive the control treatment.

**Only patients with discordant results (ie, either high clinical risk an low genomic risk or low clinical risk and high genomic risk) are randomly assigned to either the control or intervention arm.

††Patients, which are randomly assigned to the non-biomarker-based strategy arm, receive the control treatment.

‡‡Patients which are randomly assigned to reverse-based strategy receive the control treatment if they are biomarker-positive and the experimental treatment if they are biomarker-negative.

§§Study designs testing the treatment effect first in the biomarker-positive subpopulation and if the result is positive in the biomarker-negative subgroup.

¶¶Study designs testing the treatment effect in both biomarker-positive and biomarker negative subgroups simultaneously.

***Study designs testing the treatment effect in the entire study population and in the biomarker-positive subgroup separately.

†††Study designs testing the treatment effect in the overall population and in the biomarker-positive subgroup sequentially.

‡‡‡Study designs testing the treatment effect not only in the biomarker-positive and biomarker-negative subgroups but also in the entire population sequentially.

§§§PM-specific adaptive aspects could be used to stratify the patients to the treatment. Generic adaptive aspects could be considered when planning a PM trial, but they could be also found in fields outside PM.

¶¶¶A threshold is used to divide the population into 'biomarker positive' and 'biomarker negative'.

****A trial arm or clinical trial is stopped early due to pre-specified rules related to treatment efficacy and safety risk.

††††Interim analyses are pre-planned analyses, which use accumulating data in order to make an early decision or adaptation.

‡‡‡‡All patients receive the new treatment for a run-in period and then are classified as either biomarker positive or negative using a pharmacodynamics biomarker.[45]

§§§§A dynamic treatment regime consists of a sequence of individually tailored therapies during the course of a treatment.

¶¶¶¶Models to suggest optimal dosage regimes of drugs for individual patients.[46]

PK/PD, Pharmacokinetic/pharmacodynamic.

**Table 3** Trial designs classification

| Core designs | Biomarker strategy | Enrichment | Master protocols | Randomise-all |
|---|---|---|---|---|
| **Design features** | | | | |
| Framework | | | | |
| Bayesian | | | | |
| Frequentist | | | | |
| Model | | | | |
| Disease progression | | | | |
| Longitudinal | | | | |
| Hierarchical | | | | |
| Control group | | | | |
| Common/shared | | | | |
| Contemporaneous | | | | |
| Historical | | | | |
| Randomisation | | | | |
| With treatment randomisation in both biomarker-positive and biomarker-negative subgroups | | | | |
| Without treatment randomisation in the biomarker-negative subgroup | | | | |
| Only for patients with discordant clinical and genomic risk evaluation | | | | |
| Randomisation in the non-biomarker based strategy arm | | | | |
| With treatment randomisation | | | | |
| Without treatment randomisation | | | | |
| Reverse biomarker strategy | | | | |
| Subgroup specific | | | | |
| Sequential subgroup specific | | | | |
| Parallel subgroup specific | | | | |
| Biomarker positive and overall strategies | | | | |
| With sequential assessment | | | | |
| With parallel assessment | | | | |
| With fall-back analysis | | | | |
| Marker sequential test | | | | |
| Biomarker assessment | | | | |
| With biomarker assessment in the entire population | | | | |
| Without biomarker assessment in the control arm | | | | |
| Personalised medicine specific adaptive aspects | | | | |
| Adaptive enrichment | | | | |
| Adaptive signature | | | | |
| Threshold determination | | | | |
| Generic adaptive aspects | | | | |
| Adding a new arm | | | | |
| Early stopping | | | | |
| Interim analysis | | | | |
| Outcome-based adaptive randomisation | | | | |
| Sample size reassessment | | | | |
| Seamless | | | | |
| Treatment tailoring aspects | | | | |
| Pharmacodynamic biomarker assessment after run-in phase period | | | | |
| Dynamic treatment regime | | | | |
| PK/PD modelling | | | | |

PK/PD, Pharmacokinetic/pharmacodynamic.

**Table 4** General characteristics of clinical trials in personalised medicine

| Trial design | Clinical trial* n=131 (%) | Recruitment status of clinical trial as for March 2021 | | | | Disease area | | Phases | | | | | |
|---|---|---|---|---|---|---|---|---|---|---|---|---|---|
| | | Ongoing n=63 (%) | Completed n=60 (%) | nf† n=1 (%) | Unknown‡ n=7 (%) | Cancer n=113 (%) | No cancer n=18 (%) | II n=75 (%) | II/III n=13 (%) | III n=28 (%) | IV n=2 (%) | n/a§ n=12 (%) | nf† n=1 (%) |
| Adaptive biomarker design | 0 (0) | 0 (0) | 0 (0) | 0 (0) | 0 (0) | 0 (0) | 0 (0) | 0 (0) | 0 (0) | 0 (0) | 0 (0) | 0 (0) | 0 (0) |
| Adaptive parallel Simon two-stage design | 1 (0.8) | 0 (0) | 1 (1.7) | 0 (0) | 0 (0) | 1 (0.9) | 0 (0) | 1 (1.3) | 0 (0) | 0 (0) | 0 (0) | 0 (0) | 0 (0) |
| Adaptive enrichment design | 4 (3.1) | 0 (0) | 4 (6.7) | 0 (0) | 0 (0) | 0 (0) | 4 (22.2) | 0 (0) | 0 (0) | 4 (14.3) | 0 (0) | 0 (0) | 0 (0) |
| Adaptive signature design | 0 (0) | 0 (0) | 0 (0) | 0 (0) | 0 (0) | 0 (0) | 0 (0) | 0 (0) | 0 (0) | 0 (0) | 0 (0) | 0 (0) | 0 (0) |
| Adaptive strategy for biomarker with measurement error | 1 (0.8) | 1 (1.6) | 0 (0) | 0 (0) | 0 (0) | 1 (0.9) | 0 (0) | 0 (0) | 0 (0) | 0 (0) | 0 (0) | 1 (8.3) | 0 (0) |
| Basket | 35 (26.7) | 19 (30.2) | 13 (21.7) | 0 (0) | 3 (42.9) | 34 (30.1) | 1 (5.6) | 32 (42.7) | 0 (0) | 2 (7.1) | 0 (0) | 1 (8.3) | 0 (0) |
| Basket of basket design | 1 (0.8) | 1 (1.6) | 0 (0) | 0 (0) | 0 (0) | 1 (0.9) | 0 (0) | 1 (1.3) | 0 (0) | 0 (0) | 0 (0) | 0 (0) | 0 (0) |
| Biomarker strategy design with biomarker assessment in the control arm | 3 (2.3) | 0 (0) | 3 (5.0) | 0 (0) | 0 (0) | 2 (1.8) | 1 (5.6) | 0 (0) | 0 (0) | 2 (7.1) | 1 (50.0) | 0 (0) | 0 (0) |
| Biomarker strategy design with treatment randomisation in the control arm | 0 (0) | 0 (0) | 0 (0) | 0 (0) | 0 (0) | 0 (0) | 0 (0) | 0 (0) | 0 (0) | 0 (0) | 0 (0) | 0 (0) | 0 (0) |
| Biomarker strategy design without biomarker assessment in the control arm | 4 (3.1) | 2 (3.2) | 2 (3.3) | 0 (0) | 0 (0) | 0 (0) | 4 (22.2) | 0 (0) | 0 (0) | 0 (0) | 0 (0) | 4 (33.3) | 0 (0) |
| Hybrid design | 1 (0.8) | 0 (0) | 1 (1.7) | 0 (0) | 0 (0) | 1 (0.9) | 0 (0) | 0 (0) | 0 (0) | 1 (3.6) | 0 (0) | 0 (0) | 0 (0) |
| Marker stratified design | 15 (11.5) | 0 (0) | 14 (23.3) | 1 (100) | 0 (0) | 15 (13.3) | 0 (0) | 0 (0) | 0 (0) | 14 (50.0) | 0 (0) | 0 (0) | 1 (100.0) |
| Modified biomarker strategy design | 3 (2.3) | 0 (0) | 2 (3.3) | 0 (0) | 1 (14.3) | 3 (2.7) | 0 (0) | 2 (2.7) | 0 (0) | 1 (3.6) | 0 (0) | 0 (0) | 0 (0) |
| Multiarm multistage design | 7 (5.3) | 3 (4.8) | 3 (5.0) | 0 (0) | 1 (14.3) | 5 (4.4) | 2 (11.1) | 4 (5.3) | 2 (15.4) | 1 (3.6) | 0 (0) | 0 (0) | 0 (0) |
| Outcome-based adaptive randomisation design | 4 (3.1) | 2 (3.2) | 2 (3.3) | 0 (0) | 0 (0) | 3 (2.7) | 1 (5.6) | 2 (2.7) | 1 (7.7) | 1 (3.6) | 0 (0) | 0 (0) | 0 (0) |
| Platform | 18 (13.7) | 13 (20.6) | 4 (6.7) | 0 (0) | 1 (14.3) | 14 (12.4) | 4 (22.2) | 11 (14.7) | 4 (30.8) | 1 (3.6) | 1 (50.0) | 1 (8.3) | 0 (0) |
| Reverse marker biased strategy | 0 (0) | 0 (0) | 0 (0) | 0 (0) | 0 (0) | 0 (0) | 0 (0) | 0 (0) | 0 (0) | 0 (0) | 0 (0) | 0 (0) | 0 (0) |
| Sequential multiple assignment randomised trial | 1 (0.8) | 0 (0) | 1 (1.7) | 0 (0) | 0 (0) | 0 (0) | 1 (5.6) | 0 (0) | 0 (0) | 0 (0) | 0 (0) | 1 (8.3) | 0 (0) |
| Tandem two-stage design | 1 (0.8) | 0 (0) | 1 (1.7) | 0 (0) | 0 (0) | 1 (0.9) | 0 (0) | 1 (1.3) | 0 (0) | 0 (0) | 0 (0) | 0 (0) | 0 (0) |
| Umbrella | 30 (22.9) | 20 (31.7) | 9 (15.0) | 0 (0) | 1 (14.3) | 30 (26.5) | 0 (0) | 19 (25.3) | 6 (46.2) | 1 (3.6) | 0 (0) | 4 (33.3) | 0 (0) |

Continued

**Table 4** Continued

| Trial design | Clinical trial* n=131 (%) | Recruitment status of clinical trial as for March 2021 | | | | Disease area | | Phases | | | | | |
|---|---|---|---|---|---|---|---|---|---|---|---|---|---|
| | | Ongoing n=63 (%) | Completed n=60 (%) | nf† n=1 (%) | Unknown‡ n=7 (%) | Cancer n=113 (%) | No cancer n=18 (%) | II n=75 (%) | II/III n=13 (%) | III n=28 (%) | IV n=2 (%) | n/a§ n=12 (%) | nf† n=1 (%) |
| Umbrella-basket hybrid | 2 (1.5) | 2 (3.2) | 0 (0) | 0 (0) | 0 (0) | 2 (1.8) | 0 (0) | 2 (2.7) | 0 (0) | 0 (0) | 0 (0) | 0 (0) | 0 (0) |

*If the same clinical trial was labelled differently across articles, we considered the trial as example of the design reported in the paper. For instance, I-SPY 2 has been labelled as outcome-based adaptive randomisation,[15] platform[36] or umbrella design[37] and it was considered as an example for each of those trial designs.
†Not found.
‡Unknown is used to indicate a trial status that has not been verified within the past 2 years on the ClinicalTrials.gov website.
§Not applicable is used on the ClinicalTrials.gov website to describe trials without FDA-defined phases including trials of devices or behavioural interventions.
FDA, U.S. Food and Drug Administration; n/a, Not applicable; nf, Not found.

(0.8%) was not registered and seven (5.3%) presented an unknown status (meaning that the trial status has not been verified within the past 2 years on the ClinicalTrials.gov website). Out of 131, 75 (57.3%) trials were phase II studies. For four trial designs, we did not find any examples of current applications.

### Trial designs for assessing personalised versus non-personalised strategy

We identified 16 trials (16/131, 12.2%) evaluating a PM versus a non-PM strategy, which used nine different study designs (online supplemental file 1).

Three trials used a biomarker design with a biomarker assessment in the control group.[14 17 18] This study design consists of first testing the marker status of the entire study population and then randomises the patients either to a biomarker-based strategy arm or a non-biomarker strategy arm.[14] In the GILT docetaxel trial (NCT00174629), patients with advanced non-small-cell lung cancer (NSCLC) were randomly assigned to either the control arm receiving a standard therapy of docetaxel plus cisplatin or the genotypic arm in which patients with low ERCC1 levels received docetaxel plus cisplatin and those with high levels received docetaxel plus gemcitabine. In the LIFT trial (NCT02498977), liver transplant recipients were randomised to either non-biomarker-based immunosuppression (IS) weaning or a biomarker-based IS weaning. ERCC1 gene expression was assessed in patients with NSCLC, which were then randomised to either to platinum therapy or non-platinum therapy in the ERCC1 trial (NCT00801736).

Four trials used a biomarker strategy design without biomarker assessment in the control arm.[14 19–21] This design only evaluates the biomarker status in patients who are assigned to the biomarker-based strategy.[14] Patients were randomised to either the NT-pro-BNP-guided therapy or usual care in the GUIDE-IT trial (NCT01685840) and either an algorithm driven individualised haemodynamic goal-directed therapy or standard care in the iPEGASUS trial (NCT03021525). Patients with mild head injury were randomly assigned to computed tomography or observation in the hospital in the OCTOPUS trial (ISRCTN81464462) and children with a doctor's diagnosis of asthma were randomised to a PM genotype-guided treatment arm or to usual care, non-genotype-guided, control arm in the PUFFIN trial (NCT03654508).

A modified strategy design, which differs from the previous strategy designs in including multiple targeted molecular profiles,[22] was used in two trials.[22–25] Patients with refractory cancer in the SHIVA trial (NCT01771458) were randomised to receive a molecularly targeted therapy based on metastasis molecular profiling or a conventional chemotherapy. In the NCI-MPACT trial (NCT01827384), patients with an actionable mutation of interest (aMOI) were assigned to a targeted therapy based on mutation status or a therapy, chosen from the four regimes, not targeting the aMOI. We found that these two trials were

also labelled as basket trials[26–28] as well as platform trial in the case of the SHIVA trial.[29]

One trial used an adaptive strategy design for biomarkers with measurement error.[25] This design is used when a second cheaper biomarker exists and may be concordant with a more expensive one, which is considered the gold standard. This design was used with some modifications in the OPTIMA trial (ISRCTN42400492). Oestrogen receptor-positive, human epidermal growth factor receptor 2 (HER-2) negative breast cancer patients were randomised to be either in the control arm receiving the standard care (ie, chemotherapy and endocrine therapy) or in the treatment arm receiving the marker-guided therapy (ie, endocrine therapy). Patients in the treatment arm, which obtained a high-risk test, also received chemotherapy.

The Siyaphambili Study (NCT03500172) used a sequential multiple assignment randomised (SMART) design to compare an individualised intervention (ie, peer-led, individualised case management) or non-individualised intervention (ie, nurse-led mobile decentralised treatment programmes) to standard care (ie, South African standard of care) or combination of both interventions in women living with HIV.[30] The SMART design allows comparing adaptive treatment strategies, which consist of a series of tailored therapies during the course of a treatment.[31]

ProBio (NCT03903835) used an outcome-randomisation adaptive design to investigate whether a treatment based on molecular biomarker signature is more effective than standard care in men with metastatic castrate-resistant prostate cancer.

Finally, we found four trials, which evaluated a personalised versus a non-personalised strategy using a master protocol design.[32–35] IMPACT II (NCT02152254) used a basket design and UPSTREAM (NCT03088059), SAFIR02_Breast (NCT02299999) and SAFIR02_Lung (NCT02117167) an umbrella design.

### Gaps in the current research on clinical trials applied to PM

The results of this scoping review also allowed us to identify some gaps in the current research on clinical trials in PM. We identified three main gaps, which concern (1) the terminology used in labelling trial designs applied to PM, (2) the applications of complex innovative trial designs to fields outside of oncology and (3) the implementation of trials for evaluating PM strategy versus non-personalised strategy.

We found that trial designs are often labelled in different ways or mislabelled, despite this gap having been identified previously.[3 4 14 15] An example is the *Marker stratified design*, which was named using 18 different labels (online supplemental file 1). We also found that a study design adopted in a clinical trial was defined differently across the literature. For instance, the I-SPY 2 trial (NCT01042379) has been labelled as outcome-based adaptive randomisation,[15] platform[36] or umbrella design.[37] The I-SPY 2 is an ongoing platform trial, which

studies multiple therapies in the context of breast cancer in a perpetual manner with arms being added or dropped based on current knowledge and collected data. Moreover, the study design adopted in the I-SPY 2 trial includes Bayesian adaptation algorithms in order to make decisions on estimated posterior probabilities, which are calculated at frequent interim-analysis points and response-adaptive randomisation.[9] According to the new proposed classification, I-SPY 2 trial would be classified as *Master protocol* because it includes multiple substudies under the same framework, with common/shared control group, early stopping, interim analysis and outcome-based adaptive randomisation as main design features.

Moreover, another gap in the current research on PM is the lack of application of novel complex study designs to fields outside of oncology. We found that 94% (81/86) of the clinical trials which used a master protocol design were in the field of oncology.

Finally, a strong need exists for clinical trials evaluating the effectiveness of a PM strategy versus non-personalised strategy. This constitutes the third gap that we identified by mapping the evidence on clinical trials applied to PM. We found only 16 trials using nine different trial designs, which compared the two strategies.

### DISCUSSION

The present study provides a broad overview and proposes a new classification of the trial designs applied to PM.

The scoping review approach was considered to be the most suitable to respond to the extensive scope of the field. Compared with systematic reviews that aim to answer specific questions, scoping reviews are used to present a broad overview of the evidence pertaining to a topic and they are useful to examine areas that are emerging, to clarify key concepts and identify gaps.[38 39]

To our knowledge, this is the first study, which systematically reviews all trial designs, including complex innovative designs (ie, basket, umbrella and platform), applied to PM. Other systematic reviews have been performed on specific trial designs such as biomarker-guided adaptive trial designs,[15] biomarker-guided non-adaptive trials designs[14] and master protocols[8] or without considering master protocols in the search strategy.[3]

We identified 21 trial designs, 10 subtypes, and 30 variations of trial designs applied to PM, which have been classified into four core categories: *Master protocols*, *Randomise-all*, *Biomarker strategy* and *Enrichment*. *Randomise-all* encompasses the largest number of trial designs (ie, 10 trial designs, 2 subtypes and 22 variations) and *Master protocols* includes those study designs which are more frequently used in clinical trials (86/131, 65.6%). A variation of the enrichment design called *multistage adaptive biomarker-directed targeted (MAT) design*,[40] which combines some features of both targeted and adaptive designs, was included in the present review because does not present the standard characteristics of a classical enrichment design but not in our classification.

In the MAT design, biomarker-positive patients are first randomised to either treatment or standard of care and interim analyses are then conducted to monitor if the primary study objectives can be achieved.

From the different approaches applied to PM, we identified 34 central features, which were combined with the four core categories in a double-entry table. The proposed table constitutes a novel manner to classify trial designs applied to PM, considering its corresponding core category and main features (eg, PM specific or generic adaptive aspects). The classification only includes features, which are strictly related to trial designs. Methods for stratification and validation of clusters in a clinical trial (eg, data-driven subgroup identification) were considered not eligible and therefore were not included. In particular, those methods were identified and described in another recent scoping review (2021).[41] Due to the variety and diversity of trial designs currently available, this classification provides a clearer and more accessible picture of the different trial designs available in PM, helping the readers to navigate this complex field. In addition, it could be particularly helpful for researchers as a first step for understanding the different methodological approaches available for their trials.

Also, it permits to consider all the relevant features associated with a trial design reducing confusion in reporting and labelling. We believe that this classification is more accurate and appropriate for describing a trial design applied to PM in its complexity. Moreover, it could help researchers and clinicians in using a harmonised terminology for labelling a trial.

Based on the results obtained, we identified three main gaps in the current research on clinical trials applied to PM. We found that more research is needed to evaluate the efficiency of PM approach versus non-personalised standard of care. A few clinical trials (16/131, 12.2%), using nine different study designs, were found evaluating these different strategies. In addition, these trials would be particularly relevant for health technologies assessment (HTA) bodies to evaluate the incremental benefit of PM over that of non-personalised approaches, from both a clinical and economical perspective, in those situations in which a non-personalised strategy is considered standard practice. We also need more research to apply trial designs to fields outside of oncology. This last result was consistent with what was found in a recent systematic review of master protocols.[8] The review showed that the great majority of basket, umbrella and platform studies (76/83, 91.6%) were conducted in the field of oncology. In particular, no umbrella trials were found outside of oncology. Finally, in line with two previous systematic reviews,[3 4] we found that a harmonised terminology was required because it would permit increase clarity among the variety of trial designs applied to PM.

Furthermore, current applications of the identified trial designs, together with the input of some experts in the field, helped us to identify four typologies of PM. For *targeted or precision medicine*, a targeted treatment, which is specific for one disease, is identified and used to treat patients with heterogeneous diagnoses but similar disease mechanisms (eg, basket trials). *Stratified medicine* includes trials in which patients are stratified in different clusters based on the collection of data characterised by the genotype or phenotype of the individuals (eg, adaptive signature trials). The treatment is tailored to each patient in the *individualised medicine* (eg, trials using pharmacokinetic models). Finally, in *individualised medicine with a dynamic regime*, the treatment tailored to each patient is adjusted over time based on the patient's response (eg, SMART trials).

The new classification and the four typologies of PM clinical trials provide the basis for the future recommendations on the use of trial designs applied to PM and on trials assessing personalised versus non-PM strategy. These recommendations are strongly needed to conduct new studies within the context of PM and, consequently, have new direct high-quality evidence in the evaluation of co-dependent PM technologies.[42]

The present study has strengths but also limitations. This is the first scoping review, which presents an overview of all trial designs applied to PM. We followed a systematic approach to map the evidence and described the process using the PRISMA-ScR guideline. However, we restricted the search strategy to the last 15 years proving a comprehensive overview rather than an exhaustive list of trial designs used in PM. In addition, by excluding single-arm trials, which are not part of a master protocol, non-adaptive enrichment design and N-of-1 trials, we might misrepresent certain study designs used for PM. Moreover, although we conducted a pilot screening for verifying the use of the same inclusion and exclusion criteria among reviewers, we cannot exclude that we did not identify some relevant publications. The information on the definition, methodology, statistical considerations, advantages, disadvantages, utility and gaps of trial designs was extracted verbatim from the included records. However, the selection of this information could be affected by the perception of the three reviewers who conducted the data extraction. Also, even if we built on existing reviews[14 15] and carefully developed a comprehensive classification, all attempts at categorisation are reductive in nature, and different classification schemes could be proposed. We believe that all classifications are based on decisions, some of which are inevitably arbitrary. Nonetheless, our proposal allows separating between core design features that characterise the main objective of the trial and the patient flow, important aspects of the trial, and more accessory design features. It may form the basis of the evaluation of which design, and which features would be best suited for a given situation. For instance, HTA representatives could use our classification as a first step to better understand the design choice taken by the researchers and successively evaluate it.

The information extracted on the pros and cons of each approach (ie, objective 3) will be subject of further analysis and will be publish in a separate study due to

considerable volume of information collected. We will also explore the pros and cons of each approach in more detail, together with experts from academia and regulatory agencies, when preparing the recommendations on the use of trial designs applied to PM.

## CONCLUSIONS

The findings of this scoping review show that several existing trial designs are applied to PM, which can be grouped into four core categories. A new classification has been proposed that allows describing trial designs taking into account their corresponding core category and main features. It can be used by readers to explore and better understand the complex field of PM clinical trials.

**Acknowledgements** The authors thank Vanna Pistotti for her assistance with search strategy development and conduction, Ines Bouajila for collaborating to the screening process and Frank Bretz, Frank Petavy and Stephen Senn for their excellent inputs and feedback.

**Collaborators** PERMIT Group: Antonio L Andreu, Florence Bietrix, Maria del Mar Polo-de Santos, Maddalena Fratelli, Vibeke Fosse, Enrico Glaab, Rainer Girgenrath, Alexander Grundmann, Josep Maria Haro, Frank Hulstaert, Pascale Jonckheer, Setefilla Luengo Matos, Emmet McCormack, Anna Monistrol Mula, Albert Sanchez Niubo, Emanuela Oldoni, Teresa Torres.

**Contributors** Study conception and design: CG, CS, II-I, JD, LSM, LMS-G, PG, RB and RP. Methodology: CG, CS, RB and RP. Data collection and analysis: CS, FBB, MC, II-I, LSM, LMS-G and RP. Trial design classification: CS and RP. Original draft preparation: CS. Review and editing: CG, II-I, LSM, LMS-G, MC, PG, RB and RP. Responsible for the overall content as guarantor: CS. All authors read and approved the final version of the manuscript. The members of the PERMIT group were involved in the preparation or revision of the joint protocol of the four scoping reviews of the PERMIT series, attended the joint workshop (consultation exercise) or contributed to one of the other scoping reviews of the PERMIT series. PG and JD coordinate the PERMIT project. JD obtained funding.

**Funding** This project has received funding from the European Union's Horizon 2020 research and innovation programme under grant agreement No. 874825.

**Competing interests** None declared.

**Patient and public involvement** The European Patients' Forum is a member of PERMIT project. Although not directly involved in the conduction of the scoping review, they received the draft review protocol for collecting comments and feedback.

**Patient consent for publication** Not applicable.

**Ethics Approval** This study was based entirely on a scoping review of relevant published literature and did not require an ethics approval.

**Provenance and peer review** Not commissioned; externally peer reviewed.

**Data availability statement** Data are available in a public, open access repository. The data set supporting the conclusions of the research reported in this paper is available in the Zenodo repository in the PERMIT community (https://zenodo.org/communities/permit-project/?page=1&size=20). The data set can be accessed via Zenodo at https://zenodo.org/record/5874552#.Ye7wJmDEVQM with doi:10.5281/zenodo.5874552.

**ORCID iDs**
Cecilia Superchi http://orcid.org/0000-0002-5375-6018
Florie Brion Bouvier http://orcid.org/0000-0001-6364-6106
Chiara Gerardi http://orcid.org/0000-0002-2459-4769
Montserrat Carmona http://orcid.org/0000-0002-4680-7745
Lorena San Miguel http://orcid.org/0000-0002-7155-7303
Luis María Sánchez-Gómez http://orcid.org/0000-0003-4358-6032
Iñaki Imaz-Iglesia http://orcid.org/0000-0002-7864-4194
Jacques Demotes http://orcid.org/0000-0002-0807-0746
Rita Banzi http://orcid.org/0000-0002-2211-3300
Raphaël Porcher http://orcid.org/0000-0002-5277-4679

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
