## [Reviewer comments · BMJ Open]

ARTICLE DETAILS

TITLE (PROVISIONAL)	Study designs for clinical trials applied to personalised medicine: a scoping review
AUTHORS	Superchi, Cecilia; Brion Bouvier, Florie; Gerardi, Chiara; Carmona, Montserrat; San Miguel, Lorena; Sanchez-Gomez, Luis; Imaz-Iglesia, Iñaki; Garcia, Paula; Demotes, Jacques; Banzi, Rita; Porcher, Raphaël

VERSION 1 – REVIEW

REVIEWER	Lipkovich, Ilya A Eli Lilly and Company
REVIEW RETURNED	05-Jul-2021

GENERAL COMMENTS	The authors of this scoping review can be commended for processing a large amount of information and providing a number of useful summarizations of the current state of the literature. Some of the categorizations proposed by the team may be questioned. For example, the division into 4 core categories “Master protocols”, “Randomise-all”, “Biomarker strategy” and “Enrichment” may be confusing in that some designs that would often be considered 2-stage enrichment designs (with subpopulation determined after the first stage) would be classified by the authors as “Randomise-all”. Also, the 29 main features identified by the team may strike a certain reader as a bit excessive. However, I would not spend time arguing for or against any classification, as there is a great deal of arbitrariness in any such grouping, and instead focus my comments on improving the clarity of the presentation and the flow of the material, which I repeat is very useful in itself. My suggestions can be grouped in several buckets 1) Improving the flow and consistency of presentation (Intro). In the beginning of the Intro, the personalized medicine is introduced as providing therapies according to patient’s specific characteristics (page 4, lines 5-9). I suggest adding definition that was used later in Box 1 that additionally emphasizes the timing of therapy (“tailoring the right therapeutic strategy for the right person at the right time”). Indeed, focusing merely on patient’s characteristics misrepresents the scope of personalized medicine ignoring, for example, dynamic treatment regimens that use evolving patient’s outcomes to decide on multi-stage treatment strategies. In the beginning of the Intro the flow is structured in (1)-(6) segments of short sentences, which may be a leftover of the initial draft outline but seems out of place in the final report. Item (4) lists examples of complex designs (basket, umbrella, and platform), which is just 3 designs among the
---

	23 designs listed later in Table 1! The reader should be provided with a broader intro that would outline the scope of work and let the reader anticipate that s/he would encounter later. 2) Clarifying the objectives of the scoping review. As the reader can see from page 4 (lines 37-43), the team had 5 objectives. It is however not clear which of them have been accomplished. I see that objectives 1 (identifying available designs), 2 (design examples), and 5 (identifying gaps) were accomplished: the authors developed a system of 29 features and 4 core categories and identified 3 gaps. However, I do not see a discussion of pros and cons (objective 3), except that those were “extracted” from the literature. In fact, it is not clear what pros and cons of different designs may be, as clearly different designs may have quite different objectives. For example, how can one discuss pros and cons of a SMART design vs a basket design as they aim at quite different objectives. It seems, that the pros and cons can be meaningfully discussed within a subset of designs that aim at the same objectives. Also, I do not see a discussion of objective 4 (evaluation of personalized vs non-personalized strategy). 3) Related to my previous comment, it is important that the authors always distinguish the “state of affairs,” that is, how the designs are presented/described in the literature, from their own innovations and contributions. This is not always easy to see for a reader, given that a large portion of the report is based on borrowings from the literature. For example, it is clear that the last column of In Table 1 presents the 4 core categorifications invented by the team. The first column with “trial designs” apparently is the listing of all the distinct names of designs as encountered in the literature. Hence repetitions. As authors mention, the “market stratified design” appears under multiple names indicating undesirable naming heterogeneity (which is one of the identified gaps). However, it is not immediately clear for a reader whether these are indeed different names for the same thing or different design variations. It is also not clear whether column 3 labeled “Variations” is based on true variations identified by the team. Apparently not, as, for example, under “Overall strategies with fall-back analysis” we see two repeats “Fall-back design” and “Fallback design” listed. I do not see any value for repeating verbatim such obvious aliases from the literature, even if some authors use hyphenated and some unhyphenated naming. 4) Better organization and summarization of empirical material. I would see value in providing in Tables whenever possible references to the literature, for example in Table 1. I also think that Table 3 showing crossing of the core categories with 29 features would greatly benefit from filling its cells with some references. If possible, bringing information about 29 features into Table 1 (e.g by using some sort of feature abbreviation) would be useful as some designs clearly are aligned with specific feature combinations. 5) It was somewhat disappointing to me that the “design” and “analysis” features are not always clearly separated. This may mask important distinctions. For example, it seems to me very important to distinguish designs where only a very small number (often a single) pre-specified candidate subpopulations are compared with the overall population from
--	--

	those designs allowing for discovering biomarkersignatures based on a large number of candidate biomarkers using machine learning methods (such as in “Cross-validated adaptive signature design”). As a distinction between confirmatory subgroup analysis and data-driven subgroup identification/ discovery is at the core of personalized medicine (see, for example, a “Tutorial on data-driven subgroup identification and analysis in clinical trials” https://onlinelibrary.wiley.com/doi/abs/10.1002/sim.7064), I would like to see some discussion of that.
--	--

REVIEWER	Jorgensen, Andrea University of Liverpool, Health Data Science
REVIEW RETURNED	16-Sep-2021

GENERAL COMMENTS	Personalised medicine is an ever-growing field of research, and robust trial designs are essential to ensure translation into clinical practice. Your scoping review identifies many trial designs using a robust search strategy. However, I am struggling to see what this review offers over and above already published reviews on the same topic (albeit being slightly more recent). In particular, whilst there is an attempt to use new categorisations for the trial designs, it is difficult to appreciate from the paper how these new categorisations will help researchers wanting to apply the various trial designs. Undoubtedly there is a need to make the literature on the design of such trials more accessible and easier to navigate by researchers, but it is difficult to see how the current paper will achieve this. A more convincing argument as to how the paper achieves this would improve the paper greatly. More specific comments are provided below.  1. In the 'Article Summary' section the statement 'This is the first overview of all trial designs applied to personalised medicine.' is somewhat misleading since there have been many previous review of such trials, as identified in the literature search. 2. In the methods section you state "A list of trial designs, which were retrieved from two previously conducted systematic reviews (11,12), was included in the data extraction form to harmonise the names used to report the same trial design. " I am aware that often the same design is referred to using different terminology in the literature. How did you ensure that the exact same trial designs were captured under the same design category in your study ? 3. In the methods you also state that "Since many narrative reviews were already published about trial designs applied to personalised medicine, the data extraction was conducted in two phases. Firstly, two reviewers (CS, FBB) independently extracted data from the identified systematic and narrative reviews. Secondly, three reviewers (CS, FBB, MC) working independently extracted data for all the remaining selected records only if they provided new information, which was not extracted in the previous phase. " Does this mean that you included systematic and narrative reviews in your search ? If so, please ensure this is clearer from the methods section. Further, please clarify exactly what you mean by 'remaining selected records'. Does this mean any papers reporting on study designs that were not included in the reviews ? 4. It is not clear how the papers applying a particular trial design were identified - please ensure this is made clearer. It would appear sensible to have undertaken two separate searches - one for papers on methodology and one for papers applying the various designs,
--

	however it is not clear whether this was the case. 5. It is difficult to appreciate how the public and patient involvement provided feedback into the protocol. Can the authors expand on this please, as it is an interesting and novel aspect to the design of a review of methodologies such as this. 6. There is significant focus on the identification of main features and feature domains of the various trial designs, but it is unclear what methods were used to identify these and further to classify the various designs according to the features. Please ensure that further detail is provided in this regard. 7. The observation that terminology used in labelling trial designs applied to personalised medicine can be confusing is not a new observation. For example, this was highlighted previously in the papers of Antoniou et al. and therefore is not a novel finding. 8. It is highly unlikely that all trials using designs of personalised medicine have been captured within this scoping review, and therefore the conclusions in terms of previous application have limited meaning. What are your thoughts on how well these results reflect the true status of trials in personalised medicine ? 9. Much effort has been put into reclassifying the trial designs into four main categories, however I do not see the benefits of doing this. How will this categorisation help future researchers ? The same applies to the identification of design features - how does this help the research field ? It would be useful to have a more convincing justification for these aspects of the project. 10. In the discussion section you say of the classification according to features that "it may form the basis of the evaluation of which design, and which features would be best suited for a given situation." Please expand on how the features would help in this way.
--	--

VERSION 1 – AUTHOR RESPONSE

Reviewer Name: Dr. Ilya A Lipkovich

Institution: Eli Lilly and Company

Please state any competing interests or state 'None declared': None declared

We are extremely thankful for the constructive comments. Below you will find our responses to each of your points.

The authors of this scoping review can be commended for processing a large amount of information and providing a number of useful summarizations of the current state of the literature. Some of the categorizations proposed by the team may be questioned. For example, the division into 4 core categories "Master protocols", "Randomise-all", "Biomarker strategy" and "Enrichment" may be confusing in that some designs that would often be considered 2-stage enrichment designs (with subpopulation determined after the first stage) would be classified by the authors as "Randomise-all". Also, the 29 main features identified by the team may strike a certain reader as a bit excessive. However, I would not spend time arguing for or against any classification, as there is a great deal of arbitrariness in any such grouping, and instead focus my comments on improving the clarity of the presentation and the flow of the material, which I repeat is very useful in itself.

The example provided by the reviewer shows how our categorization may be confusing in terms of wording for some readers. As rightly stressed out by the reviewer, a 2-stage enrichment design would be included in the *Randomize-all* category, according to our categorization, because all patients would be randomised to either experimental or control groups during the first stage. However, we believe that this 'confusion' in wording is limited to a few trial

designs and can be avoided by carefully reading the definitions of the categories provided in the manuscript.

As we acknowledged in the discussion section (lines 44-45, page 10), “*all attempts at categorisation are reductive in nature, and different classification schemes could be proposed.*” All things considered, we believe that our categorization is a novel attempt to classify the many different trial designs applied to the field of personalised medicine, and can be helpful for readers to navigate in the complex field of personalised medicine trials, particularly by helping investigators understand the different design options that exist for their trials. We have therefore re-emphasized the point highlighted by the reviewer about the arbitrariness of any classification.

Lines 43-45, page 10

“Also, even if we built on existing reviews (14,15) and carefully developed a comprehensive classification, all attempts at categorisation are reductive in nature, and different classification schemes could be proposed. We believe that all classifications are based on decisions, some of which are inevitably arbitrary.”

My suggestions can be grouped in several buckets

- 1) **Improving the flow and consistency of presentation (Intro). In the beginning of the Intro, the personalized medicine is introduced as providing therapies according to patient’s specific characteristics (page 4, lines 5-9). I suggest adding definition that was used later in Box 1 that additionally emphasizes the timing of therapy (“tailoring the right therapeutic strategy for the right person at the right time”). Indeed, focusing merely on patient’s characteristics misrepresents the scope of personalized medicine ignoring, for example, dynamic treatment regimens that use evolving patient’s outcomes to decide on multi-stage treatment strategies.**

In the introduction, we have added the definition of personalised medicine provided in Box 1.

Lines 2-8, page 3

“Personalised medicine is an evolving field, which allows treating patients by providing them a specific therapy according to their individual demographic, genomic or biological characteristics (3). It was defined by the European Council Conclusion on personalised medicine as ‘a medical model using characterisation of individuals’ phenotypes and genotypes (e.g. molecular profiling, medical imaging, lifestyle data) for tailoring the right therapeutic strategy for the right person at the right time, and/or to determine the predisposition to disease and/or to deliver timely and targeted prevention’ (4).”

In the beginning of the Intro the flow is structured in (1)-(6) segments of short sentences, which may be a leftover of the initial draft outline but seems out of place in the final report. Item (4) lists examples of complex designs (basket, umbrella, and platform), which is just 3 designs among the 23 designs listed later in Table 1! Thereader should be provided with a broader intro that would outline the scope of work and let the reader anticipate that s/he would encounter later.

We have reviewed the flow of the introduction and have better highlighted the scope of work.

Lines 10-40, page 3

“Many trial designs have been used to evaluate personalised treatment or interventions (3). The most common design is the enrichment design, whereby only biomarker positive patients are randomly assigned to the targeted or control arm (4). Despite its popularity, the use of enrichment designs is recommended only when the biomarker is a perfect predictor of the response in order not to deny biomarker-negative patients a treatment they would have otherwise benefited from (5). Prospective validation of the candidate biomarker is therefore strongly recommended before applying these trials designs.

Over the last years, more complex study designs have been increasingly proposed in the field of personalised medicine (4). According to the Clinical Trials Facilitation and Coordination Group, a clinical trial is considered as using a complex design “if it has separate parts that could constitute individual clinical trials and/or is characterised by extensive prospective adaptations such as planned additions of new Investigational Medicinal Products (IMPs) or new target populations” (6). These designs are particularly efficient because allow answering multiple clinical research questions within a single study (7). Examples of common complex designs are the so-called basket, umbrella, and platform trials, which are frequently applied in the field of oncology (8). Basket trials refer to designs in which patients with heterogeneous diagnoses but with similar disease mechanisms are tested using the same targeted therapy. While, umbrella trials evaluate multiple treatment options in patient groups, which present the same disease, but with different genetic mutations. Finally, platform trials allow testing multiple targeted therapies in patients with the same disease in a perpetual manner, using interim evaluations and allowing therapies to enter or leave the trial (9). However, these designs are often challenging (6) because they often require independent statistical analyses for each sub-protocol, including interim analyses driving prospective adaptation with the addition of new interventions or populations, and/or termination of sub-protocols based on futility or safety issues.

Numerous methodological challenges, covering many aspects of the study design (e.g., randomization, use of control arm, biomarker stratification, biomarker validation), are associated with trial designs applied to personalised medicine. The application of robust methodologies is especially important for clinical trials applied to personalised medicine to correctly select participants and treatments to be tested. As a starting point for the development of new recommendations on the use of trial designs applied to personalised medicine, we aimed to map the landscape of the existing study designs for clinical trials applied to this medical field.”

- 2) **Clarifying the objectives of the scoping review. As the reader can see from page 4 (lines 37-43), the team had 5 objectives. It is however not clear which of them have been accomplished. I see that objectives 1 (identifying available designs), 2 (design examples), and 5 (identifying gaps) were accomplished: the authors developed a system of 29 features and 4 core categories and identified 3 gaps. However, I do not see a discussion of pros and cons (objective 3), except that those were “extracted” from the literature. In fact, it is not clear what pros and cons of different designs may be, as clearly different designs may have quite different objectives. For example, how can one discuss pros and cons of a SMART design vs a basket design as they aim at quite different objectives. It seems, that the pros and cons can be meaningfully discussed within a subset of designs that aim at the same objectives.**

As we reported in lines 36-38, page 4, as well as in lines 52-53, page 10, the objective 3 was not addressed in this paper and was subjected to a specific ongoing study. Therefore, no results and discussion related to objective 3 is reported in the manuscript. As the reviewer correctly highlighted in his comment, we believe that identifying and describing, and consequently comparing, pros and cons of different trial designs with (sometimes) very different objectives is not informative, and recognize that this wording can be misleading. Therefore, we rather intended to focus on the advantages and limitations of the most commonly used study designs in the field of personalised medicine (in particular, adaptive enrichment/adaptive signature, umbrella and basket trial design). Future steps in the project will focus on providing guidance for choosing a design in a specific situation.

Also, I do not see a discussion of objective 4 (evaluation of personalized vs non-personalized strategy).

Thank you for raising this issue. We have added a discussion on objective 4.

Lines 2-8 page 10

“We found that more research is needed to evaluate the efficiency of personalised medicine approach vs. non-personalised standard of care. A few clinical trials (16/132, 12.1%), using nine different study designs, were found evaluating these different strategies. In addition, these trials would be particularly relevant for Health Technologies Assessment (HTA) bodies to evaluate the incremental benefit of personalised medicine over that of non-personalised approaches, from both a clinical and economic perspective, in those situations in which a non-personalised strategy is standard practice.”

3) **Related to my previous comment, it is important that the authors always distinguish the “state of affairs,” that is, how the designs are presented/described in the literature, from their own innovations and contributions. This is not always easy to see for a reader, given that a large portion of the report is based on borrowings from the literature. For example, it is clear that the last column of In Table 1 presents the 4 core categorifications invented by the team. The first column with “trial designs” apparently is the listing of all the distinct names of designs as encountered in the literature. Hence repetitions. As authors mention, the “market stratified design” appears under multiple names indicating undesirable naming heterogeneity (which is one of the identified gaps). However, it is not immediately clear for a reader whether these are indeed different names for the same thing or different design variations.**

VERSION 2 – REVIEW

REVIEWER	Lipkovich, Ilya A Eli Lilly and Company
REVIEW RETURNED	01-Feb-2022
GENERAL COMMENTS	All my previous comments have been addressed. I have no additional comments.
REVIEWER	Jorgensen, Andrea University of Liverpool, Health Data Science
REVIEW RETURNED	01-Feb-2022
GENERAL COMMENTS	Thank you for addressing my previous comments, I'm glad you found them useful.